# Exploring the Impact of Sustainability Trade-Offs: The Role of Product and Sustainability Types in Consumer Purchases Mediated by Moral Regulation

**DOI:** 10.3390/bs14080702

**Published:** 2024-08-12

**Authors:** Munshik Suh, Je Eun Yoo

**Affiliations:** 1Business Administration, Pusan National University, Busan 46241, Republic of Korea; msisuh@pusan.ac.kr; 2Global Business Administration, Kyungsung University, Busan 48434, Republic of Korea

**Keywords:** sustainability, sustainability trade-offs, moral regulation, product attribute, sustainability type, sustainability attitude–behavior gap

## Abstract

The attitude–behavior gap in sustainable product purchasing persists despite prior attempts to address it, thus indicating a need for more research. This study examines contextual factors in sustainable consumption, particularly the sustainability trade-offs (STOs) faced by consumers during product purchases and the impact of conditional morality. We investigate STOs in terms of sustainability type and attribute type and indicate that moral regulation enhances the impact of sustainability trade-offs on purchase intention. Four experiments were conducted with 457 participants in South Korea, focusing on STOs in terms of sustainability type (social vs. environmental) and attribute type (utilitarian vs. hedonic). The findings indicate that attitudes towards sustainability have a positive influence on purchase intention, though sustainability- and attribute-type STOs do not significantly affect this relationship. However, the combination of STOs in environmental sustainability with utilitarian attributes does have a significant impact on the relationship between attitude and purchase intention. Furthermore, while moral regulation mediates this relationship, morality does not have an impact. This research highlights the moderating role of sustainability trade-offs in the relationship between attitudes towards sustainability and purchase intention, underscoring the importance of contextual factors in sustainable product purchasing. Firms can leverage sustainability trade-offs in their marketing strategies, incorporating product features and advertising messages.

## 1. Introduction

Sustainable consumption is on the rise and represents a significant trend, resonating with consumers due to shifting societal norms and a greater understanding of environmental and social challenges [1]. Scholars have observed an increasing recognition of these concerns within the market, leading to more favorable attitudes towards sustainable goods and a heightened emphasis on personal responsibility [2,3]. As a result, firms are progressively pursuing sustainability goals through various approaches, such as incorporating eco-friendly production processes and introducing innovative sustainable products targeting new market segments.

The integration of the Environmental, Social, and Governance criteria as a framework for assessing corporate performance has heightened the emphasis on sustainability within organizations. This has encouraged firms to modify their offerings to align with sustainable practices. The implementation of sustainability initiatives may be mandated through regulations on both governmental and corporate levels, while at the consumer level, companies are integrating sustainable features into their products [4]. Despite a significant number of consumers claiming to be environmentally and socially conscious and indicating a readiness to alter their consumption patterns, the market share of sustainable products remains modest, hovering around 3–4% [5]. This phenomenon is called the ‘sustainability attitude–behavior gap’, which refers to the discrepancy between a consumer’s pro-environmental and social attitude and their actual purchasing behaviors [6,7,8,9,10,11,12,13]. Numerous studies have been conducted in order to address the gap by exploring the trade-offs among various dimensions of sustainability domain [14], supply chains [15], consumers’ personal characteristics, environmental education [15], trust mechanisms [16], and other factors. 

Theoretical frameworks such as Ajzen’s Theory of Planned Behavior [17] and Stern’s value–belief–norm theory propose that personal attitudes and norms significantly influence sustainable consumption [3,17,18]. These theories suggest that assuming responsibility for the consequences of consumption places decisions within a moral framework, indicating that sustainable choices are driven by attitudes shaped by morality, environmental and social concerns, and sustainable consciousness [9,19,20,21,22,23,24]. However, there is some contention regarding whether focusing solely on intrapsychic factors can fully elucidate the disparity between attitudes and behaviors in sustainable practices [9,12,25,26]. This underscores the necessity to consider psychological and contextual factors as well as intrapsychic factors, particularly within the purchase context.

Emphasizing the gap is essential for developing interventions that can bridge the discrepancy; however, a significant amount of research has been directed towards the psychological aspects of consumers [12]. Consumers with positive environmental attitudes may still behave differently based on context, and the trade-offs and conflicts relating to personal over altruistic gain are more common than exceptional [6,27]. Therefore, as the purchase of a product is influenced by numerous situational factors, further exploration focusing on the contextual elements in sustainable consumption is needed. Expanding on the research of various academics [12,28,29,30,31,32], this investigation posits that individual traits and the purchase context, specifically the types of sustainability and product attributes, can impact the purchase of sustainable products [3,17,18].

In sustainable consumption, attitude does not solely determine an environmentally friendly purchase [33]. Product attributes play a key role in sustainable product choices [34,35,36,37], leading to inter-attribute trade-offs [29,38,39,40]. Sustainable products often necessitate trade-offs for users, such as increased prices, diminished quality, or reduced performance, resulting in a value–action gap. Despite the significance of trade-offs in enhancing sustainable consumption, most studies on sustainability trade-offs (STOs) concentrate on governmental and corporate spheres as regulatory measures or solely on the environmental aspect of sustainability. Consequently, a more detailed examination is imperative within this domain.

Prior research focuses on sustainability trade-offs (STOs) in two aspects. Social sustainability has been less clearly defined in contrast to environmental sustainability and often disregarded in sustainability research. Consumer perceptions of each sustainability type vary [41], influencing decision making on trade-offs. Emotional factors play a role in decision making regarding inter-attribute trade-offs [42]. Also, some researches posit on the concept of the ‘sustainability liability effect’, where sustainable attributes are seen as weak [43,44,45]. Conversely, others suggest a spillover effect in sustainable consumption due to minimal individual effort [46,47]. In ethical purchasing, consumers intended to purchase the product motivated from the product desirability to justification of the purchase [48,49]. STOs result in decisions that are perceived as ‘less guilty’ or easily justifiable; therefore, opting for utilitarian features can reduce guilt compared to hedonic features. Consumers make trade-offs between sustainability and utilitarian or hedonic qualities, resulting in varied emotions and purchase intentions. However, sustainability trade-offs (STOs) at the product level become particularly crucial, especially in cases where sustainability is integrated as a product characteristic, resulting in direct trade-offs when consumer choices are constrained. Further research is warranted to explore inter-attribute trade-offs in sustainable product procurement with the aim of narrowing the gap between attitudes and behaviors.

Morality serves as a significant motivation for sustainable consumption because it aligns with ethical principles and personal values that drive individuals to act in ways that benefit society and the environment. Some researchers argue that moral motivations such as moral norms [43,50,51,52,53,54], personal norms, perceived self-efficacy [55,56], sustainable consciousness, and perceived effectiveness [22,57] are crucial for fostering sustainable consumption practices. However, the decision-making process is complex and influenced by many factors, suggesting that sustainability may not be viewed as a moral imperative [58,59,60,61,62], and a social dilemma exists in sustainable consumption [58,63]. Some studies argue that morality of consumer may be a psychological mechanism underlying sustainable product purchasing [64,65,66,67]. Moral regulation involves psychological processes in balancing moral and immoral actions for a sense of equilibrium, referring to the internal and external mechanisms that guide consumers to follow their moral standards [59,67,68,69,70]. Moral regulation includes moral licensing and moral cleansing [59,71,72,73] and extends to situations in which individuals engage in moral behavior to restore self-worth after taking negative actions.

In the context of STOs, consumers might choose options based on their self-justification rather than just their attitudes towards sustainability (ATS), which are mainly rooted in one’s morality; they may also be influenced by moral licensing or cleansing, which impact consumer decision making. The reasons behind the gap between attitude and behavior in sustainability, which ultimately deters consumers from making purchases, include consumers prioritizing self-interest and contextual factors such as STOs. Consequently, there is a need to understand the mechanism that drives the selection of conflicting ideas. Throughout our research, we aim to elucidate several research inquiries, as follows:RQ1. How does the consumer’s attitude towards sustainability influence their tendency to purchase sustainable products?RQ2. In the realm of sustainable product purchasing, do the contextual factors besides the morality of consumer impact STOs (sustainability trade-offs)? Specifically, do the types of sustainability and the attributes of the product hold substantial importance within the STO framework?RQ3. What are the underlying psychological mechanisms in STOs and how do they impact the consumer’s decision to purchase sustainable products?

## 2. Literature Review and Hypothesis

### 2.1. Consumer’s Attitudes towards Sustainability and the Attitude–Behavior Gap in Sustainability

The World Commission on Environment and Development (WCED, 1987) [74] proclaimed the importance of ‘sustainability’ and noted environmental integrity, social equity, and economic prosperity as its three pillars. Achieving and maintaining sustainability has evolved into a widely acknowledged norm in society, being an aspect of the expected ethical conduct observed in consumer behavior, thus fostering a collective consensus on the commitment to sustainable practices [1]. Since the Environmental, Social, and Governance (ESG) standards are executed as an evaluation tool for firms, the corporate world has led an increased focus on sustainability at the organizational level, modifying their services and product offerings to align with more eco-friendly practices [75].

The attitude–behavior model is the most utilized one for understanding consumption behaviors [76,77], and studies have argued that consumers who are conscious about environmental and social issues are likely to show sustainable behavior [78,79,80,81]. In addition, a consumer’s pro-social status significantly affects their intentions for sustainable behavior [82]. Altruism, perceived consumer effectiveness, happiness, and status enhancement are established as factors that affect customers’ green purchase intentions positively by mediating their attitude [83]. The theory of environmental consciousness posits that environmental awareness and perceived effectiveness are the main indirect influencers of green consumption [22]. Therefore, consumers with a higher concern for environmental and social issues are likely to have more positive and stronger attitudes towards sustainability and are likely to demonstrate sustainable consumption.

Attitudes towards sustainability are often shaped by the need to navigate and manage trade-offs between various sustainability dimensions, such as economic, environmental, and social aspects. The concept of attitude towards sustainability (ATS) pertains to the level of importance consumers place on sustainability and how they prioritize it when making purchases [39]. This mindset encompasses various considerations, including the social, environmental, and ethical aspects of sustainability. ATS plays a significant role in how consumers weigh the trade-offs between sustainability and other attributes of a product, such as hedonic (e.g., esthetics) and utilitarian (e.g., functional performance) values, impacting their emotional responses and decision-making processes.

Environmental concerns are important in the shift towards sustainable goods, but a gap between consumer attitudes and actions towards sustainability remains [84]. Cowe and Williams (2000) highlighted that even though 30 percent of consumers demonstrated an inclination to buy eco-friendly items, these purchases represented less than 3 percent of the overall market share [5]. This disparity is widely recognized in the academic literature and is commonly known as ‘the green attitude–behavior gap’ [8], ‘the green intention–behavior gap’ [25], or ‘the motivation–behavior gap’ [84]. Currently, numerous studies are dedicated to elucidating, comprehending, and addressing this phenomenon [7,12,26]. Elhaffar et al. [12] argued that research focused on the motivations behind and barriers to sustainable consumption broadly lacks focus on the green gap phenomenon and the methodologies used in addressing it. Most academic research utilizes the economic rational paradigm to examine the green gap phenomenon primarily through the Theory of Planned Behavior (TPB), which enforces a specific alignment between attitudes, subjective norms, and perceived control. In contrast to conventional purchase behavior, where the binary options of ‘purchase’ or ‘non-purchase’ exist, in sustainable purchase behavior, the consumer categories become more varied, including ‘green gapper’ between ‘non-green consumer’ and ‘green consumer’ [12]. Though consumer attitudes, norms, and other intrapsychic factors are the not the only determinants of sustainable behavior [33,85], contextual elements serve as significant variables impacting the sustainability gap, influencing it either in a positive or negative manner and exerting an effect on the consumer’s purchase intention (PI) on sustainable products.

Not only do one’s personal norms and altruism affect PIs for sustainable products; other contextual factors also impact on PI for the sustainable product. That is, purchasing of sustainable products often necessitates trade-offs for users, such as increased prices, diminished quality, or reduced performance, resulting in an attitude–behavior gap. Considering the trade-offs, the scope of sustainability in social, environmental, and economic dimensions [31,85,86] can be applied as an attribute. Consumers undergo a psychological process of trading off sustainability within the product against other attributes in evaluation [85]. With regard to previous research [39], this study defines the term sustainability trade-off (STO) as ‘the exchange in the ratio of the product attribute (A) that planned to provide the intrinsic value to the consumer with the attribute of sustainability (B)’. The approach does not entail selecting one characteristic over the other but rather the extent to which they are factored in with a consistent total value of 10, indicating that the overall value stays constant regardless of the changing proportion between sustainability and product features. Examples of STOs are shown in Figure 1.

Since not only consumer attitudes towards sustainability but also contextual factors affect the sustainable product purchase intention, we first investigate the relationship between a customer’s attitude towards sustainability and its impact on purchase intentions (PIs) of the sustainable product, and then we will further look into the effect of contextual factors affecting PIs. Therefore, we can posit that the attributes of sustainable products, such as eco-labels, the types of attributes applied as green, and eco-packaging, play a crucial role in enhancing consumers’ purchase intentions (PIs). Therefore, we posit the following hypothesis:
**H1.** *Attitudes towards environmental and social sustainability will have a positive effect on sustainable product purchasing*.

#### 2.1.1. Sustainability-Type Sustainability Trade-Offs (STOs): Environmental vs. Social

In reference to the Brundtland Report in 1987 [5], sustainable development is commonly defined through three key dimensions: environmental, economic, and social. Social sustainability is commonly defined as an enriching state of being within societies and a mechanism within societies capable of attaining such a state [87]. Since the inception of the three-pillar sustainable development, social sustainability has consistently remained the most ambiguous and least-defined pillar [88]. Social sustainability is often seen as less important compared to economic and environmental sustainability [89,90]. Consumer perception of each sustainability pillar varies, as consumers view environmental sustainability as cognitive, long-lasting, and global while they regard social sustainability as emotional, short-term, and more specific [41]. These varying perspectives on environmental and social sustainability can significantly impact the decision-making process regarding trade-offs in sustainability. Hanson et al. (2019) contended that the level of tangibility could distinguish the influence of social/environmental sustainability on the formation of brand attitudes [91]. This implies that when companies establish their brand identity through physical attributes or products, creating the perception of a product as tangible and visible and as having a natural material composition [92], consumers establish a stronger link with environmental sustainability. That is, the tangible characteristics of a product are likely to exert a stronger impact on consumers’ decision-making processes when they are faced with environmental STOs. Therefore, the impact of environmental STOs on customer purchase intentions will depend on the sustainability type, so we propose the following hypothesis.

**H2.** *Sustainability-type STOs will moderate the effect of ATS on PI*.

#### 2.1.2. Attribute-Type Sustainability Trade-Offs (STOs): Utilitarian vs. Hedonic

Prior research has examined how sustainability affects product perception, indicating that sustainability is often viewed negatively in terms of functionality and strength. This perception leads consumers to seeing sustainable products as having an inferior level of performance [93,94,95]. The concept of ‘sustainability liability’ underpins consumer beliefs about sustainability reducing product effectiveness and leading to lower performance in sustainable products. Some studies propose a connection between sustainability, STOs, and regulatory focus; for instance, promotion-focused consumers might be more responsive to positive framing and the potential benefits of sustainable products, while prevention-focused consumers might respond better to information emphasizing the avoidance of negative outcomes. Additionally, the social dilemma of sustainable consumption, where individuals recognize the collective benefits but are deterred by higher costs, can be mitigated by leveraging the concept of conditional cooperation. This approach can be particularly effective if consumers are convinced that others are also making sustainable choices, thus aligning with their regulatory focus and reducing the moral dilemma they face [38,96,97]. Since marketing mix factors and societal variables are utilized to explain the attitude–behavior gap in sustainability [98], we focus on the trade-offs in sustainability and the contextual factors with products. A study in consumer behavior science used neutralization techniques to explain norm-breaking behaviors and later applied them to ethical and sustainable behaviors [99], while purchase decisions have been shown to be based on desirability perceptions, but ethical considerations introduce the factor of ‘justifiability’ [48,49].

Previous research suggests that choosing utilitarian over hedonic features can lead to consumers experiencing less feelings of guilt [78]. Consumers tend to justify their decisions based on utilitarian rather than hedonic attributes; therefore, opting for utilitarian features is viewed as a morally better choice than hedonic features, and consumers may feel less guilty prioritizing utilitarian attributes over sustainability.

**H3.** *Attribute-type STOs will moderate the effect of ATS on PI*.

**H3a.** *Trade-offs with utilitarian attributes will negatively affect the influence of ATS on PI*.

**H3b.** *Trade-offs with hedonic attributes will positively affect the influence of ATS on PI*.

#### 2.1.3. The Interaction Effect of Sustainability- and Attribute-Type STOs

Scholars have noted that one reason for the sustainability attitude–behavior gap is the limited and separate research in intrapsychic factor and contextual factors in STOs, thus calling for more investigation from the holistic perspective [12,99]. To address this, we propose combining intrapsychic factors such as ATS and contextual factors such as sustainability- and attribute-type STOs. Environmental sustainability is expected to have a more significant impact on STOs as consumers strongly associate the environment with the product [90,100]. Furthermore, individuals are inclined towards selecting sustainable goods to relieve feelings of guilt, thus facilitating the justification of opting for a non-sustainable alternative based on specific characteristics. Therefore, the combination of environmental STOs, STOs with the utilitarian attribute, and ATS is likely to have a stronger impact on the effect of ATS on PI.

**H4.** *Sustainability- and attribute-type STOs will moderate the effect of ATS on PI*.

### 2.2. Sustainability Trade-Offs and Moral Regulation

Moral regulation refers to a process whereby individuals strive to maintain an optimal level of moral self-worth over time by balancing past actions with subsequent behaviors. This regulation involves moral cleansing, where individuals engage in moral behavior to offset prior immoral actions, and moral licensing, where past good deeds lead to less moral behavior due to a positive self-perception [100]. The relationship between moral regulation, moral cleansing, and moral licensing is characterized by a dynamic pattern of self-regulation, where individuals alternate between altruistic and selfish behaviors to maintain their perceived moral self-worth. Moral regulation includes moral licensing and moral cleansing [59,71,72,73], which extends to situations where consumers engage in moral behavior to restore self-worth after committing negative deeds. Moral regulations have received significant attention in the literature, with a focus on areas such as racial prejudice [69], purchasing behavior [101], and donations [102]. However, there is a lack of studies investigating these phenomena in the context of environmental action. Scholars have argued that engaging in green behaviors could reduce subsequent cooperative actions, with individuals who purchased from a green store being more likely to engage in stealing and cheating in a following task compared to those who bought conventional products [101]. Studies on ‘moral cleansing’ suggest that individuals may try to balance their negative actions by engaging in behaviors that symbolically cleanse them of past transgressions and reaffirm their moral self-regard. Sustainable behavior can be demonstrated through pro-social, pro-environmental [33], and altruistic actions [18,24] that genuinely have a moral basis [62,103,104].

Various scholars have highlighted a positive connection between personal values and pro-environmental conduct, with an individual’s moral norms and values serving as significant motivators [3,105]. Nevertheless, scholars have also observed that focusing on the morality of consumers, such as their biospheric and altruistic values, may not provide a comprehensive explanation for sustainable actions when it comes to ‘purchase matter’ [105,106]. One reason for the attitude–behavior gap in sustainability which eventually leads to a consumer not to making a purchase is consumers prioritizing their self-interest and contextual factors such as STOs [12,39]. Sustainable living is seen as the societal norm in the current market, and not following this norm may lead individuals to neglect their ethical responsibilities and experiencing feelings of remorse. Consumers exhibit diverse ethical standards, with a focus on personal decisions over universal morality. Participating in sustainable actions can create ‘moral credit’, potentially reducing the drive for additional moral behaviors. Therefore, we argue that consumers tend to align their moral actions with their individual moral compass in sustainable consumption.

When consumers are confronted with a purchasing context, they are inevitably confronted with STOs and must compromise between their self-interest and biospheric values. Consumers value ‘justifiability’ and ‘avoiding guilt’ in decision making, and when it comes to STOs, consumers are likely to make a moral decision based on their moral regulation; that is, how guilty they may feel during the STO will influence their sustainable purchase behavior. Therefore, we propose the following hypothesis:
**H5.** *In an STO condition, moral regulation will mediate the effects of environmental concern on purchase intention*.

## 3. Materials and Methods

### 3.1. Pre-Test

A pre-test was conducted to evaluate stimuli suitability for sustainability domains among university students and adults in Busan and Ulsan, South Korea, over three days in April 2022. Social and environmental sustainability were linked to United Nations Sustainable Development Goals. Participants associated social sustainability with aspects like poverty alleviation, human rights, and service to the community and environmental sustainability with actions like climate change mitigation, green consumption, etc. Participants further connected environmental sustainability with practices such as using recycled materials and avoiding animal testing, while social sustainability was linked to community welfare and fair trade. The study explored participants’ perceptions of sustainability dimensions based on affective vs. cognitive, short-term vs. long-term, global vs. local, and tangible vs. intangible aspects. Questionnaires were adapted from Catlin et al.’s (2017) research [41]. The results show each distinct perception of sustainability can classified as affective (M_social_ = 1.50 vs. M_environ_ = 6.25, *t* = 0.754, *p* < 0.001), short-term (M_social_ = 1.97 vs. M_environ_ = 0.816, *t* = −9.050, *p* < 0.001), global (M_social_ = 1.94 vs. M_environ_ = 6.29, *t* = −15.312, *p* < 0.001), and tangible (M_social_ = 2.64 vs. M_environ_ = 6.42, *t* = −15.312, *p* < 0.001). Our study investigated consumer sustainability trade-offs in product attributes, specifically examining whether consumers, when facing an attitude–behavior gap with respect to sustainability, tend to prioritize certain product attributes. We propose that this gap manifests differently depending on whether the product attribute is hedonic or utilitarian.

Initially, we conducted the pre-test to assess consumer perceptions of the coffee pot as a product that exhibits neither purely hedonic nor utilitarian characteristics. A paired-sample *t*-test was revealed for a laptop, gaming console, and coffee pot. In the survey, products rated 4 or below were classified as utilitarian, while those rated above 4 were classified as hedonic. The laptop, with a mean rating of 2.33, was considered a utilitarian product, whereas the gaming console, with a mean rating of 5.75, was considered a hedonic product. However, for the coffee pot, no significant differences in consumer perception of the product as utilitarian or hedonic were observed. The results are shown as utilitarian (M_utilitarin_ = 3.96) and hedonic (M_hedonic_ = 4.14) attributes of the coffee pot (*t* = 3.08, *p* > 0.06), thus validating our selection of the coffee pot as a suitable experimental stimulus.

For the main experiment, we aimed to observe how consumers perform sustainability trade-offs with the attributes of the coffee pot. We hypothesized that the attitude–behavior gap towards sustainability would influence consumers’ prioritization of hedonic versus utilitarian attributes in different ways.

By exploring these dynamics, our research contributes to understanding consumer behavior in sustainable product choices, particularly in contexts where products may not clearly align with traditional categorizations of hedonic or utilitarian attributes.

### 3.2. Participants and Procedure

An open-ended questionnaire was conducted with 199 South Korean participants from Busan National University and Ulsan College. A follow-up survey was conducted with 260 participants through an online research institution (http://surveybilly.com, accessed on 21~28 April 2022). A total of 457 responses out of 459 were used for the analysis, and the details of the participants are shown in Table 1.

Participants were first asked about their attitude to sustainability via statements we modified from a prior study [39], such as “*Issues of social sustainability are important to me (e.g., labor safety, fair labor, community contribution)*”, “*Environmental sustainability issues are important to me (e.g., recycling, energy efficiency, minimizing pollution)*”, and “*It is important to me that companies hold high ethical standards*” as for whole sustainability. All the scales were measured on a 7-point Likert scale. Then, participants read the scenario with the option to select between two coffee pots that were identified as neutral in the pre-test, differing in terms of trade-off type (hedonic vs. utilitarian) and sustainability type (social S vs. environmental S). The methodology followed previous practices by defining utilitarian value as the functional performance of the product and hedonic value as the esthetics of the product [39,96].

Initially, the two experiments were divided into two sustainability types (social vs. environmental). Subsequently, for each type of STO, two experimental scenarios were implemented for the STO with product attributes (utilitarian vs. hedonic), creating four kinds of STO scenarios (social vs. environmental sustainability, and utilitarian vs. hedonic attribute) with a between-subject design. For the sustainability type, we described social sustainability as *‘the coffee pot company opens coffee lecture for the community’*, *‘the firm has strong policy to keep the labor right’*, and *’the firm donates for the fair trade in coffee beans’*. For environmental sustainability, we described the coffee pot as being made of *‘recycled materials’* and *‘reusable plastics’*, and there being a *‘reduction in the pollution on the production process’* (the items were modified from a previous study [38]). In the STO with product attributes, we described the coffee pot as having won an award from the ‘Consumer Report 2022’ for its ‘*design* (hedonic feature)’ and its ‘*performance* (utilitarian feature)’, as modified from prior research [39]. For each scenario, the STO was described as noted in Figure 1. However, we set the higher weight of the attribute as ‘9’ and the lower weight as ‘5’ for the participant to perceive that each attribute works because they were over half of the total value. The example of the scenario is shown in Figure 2.

After reading the scenario, participants were asked which sustainability type was applied in the scenario according to a 7-point Likert Scale ranging from 1 (very social) to 7 (very environmental). Participants were then asked to select the coffee pot they preferred and rate their purchase intention according to a 7-point Likert Scale ranging from 1 (do not agree) to 7 (totally agree). To minimize bias, participants were briefed on sustainability concepts and the survey purpose before responding, with projective techniques used to gauge attitudes and ethical self-perception regarding sustainability. Finally, participants were asked to rate their level of agreement on moral regulation based on the following three statements (modified from previous research [69,107]) on a scale ranging from 1 (strongly disagree) to 7 (strongly agree): *‘the choice I made was consistent with my moral standard’*, *‘I believe that by making this choice, I have made sufficient effort to achieve my moral goals’*, and *‘I believe that by making this choice, I have made sufficient effort to achieve my moral goals’*. Subsequently, participants were assessed on their moral identity using the Moral Identity Scale developed by Aquino and Reed (2002) [108].

## 4. Results

### 4.1. Measurement Reliability and Validity

Reliability, as evaluated through Cronbach’s alpha and composite reliability (CR) coefficients, demonstrates a high level of consistency for nearly all constructs, with values surpassing the threshold of 0.70 [109]; therefore, the factor loadings for moral regulation item 3 and morality items 4 and 5 were below 0.7 and were therefore deleted. A summary of the reliability and validity analysis is presented in Table 2.

### 4.2. The Results of Hypothesis Testing

The relationship between attitudes and sustainability (ATS) and purchase intention (PI).

The aim of this study is to examine the relationship between consumers’ attitudes towards sustainability and their purchase intentions, as well as the various contextual factors that influence these intentions. Considering that sustainability involves social and environmental elements [110], this investigation examines how attitudes towards each of these dimensions, as well as the overarching attitude towards sustainability, impact purchase intention. In this study, three categories of attitudes towards sustainability (ATS) were quantified, specifically environmental, social, and holistic ATS, followed by an examination of their influence on the intent to purchase sustainable goods. We executed a multiple linear regression analysis via the software package SPSS ver.29 to evaluate how the distinct ATS can predict the purchase intention for sustainable products. Neither the attitude towards environmental sustainability (**β** = 0.076, *p* > 0.005) nor the attitude towards social sustainability (β = 0.032, *p* > 0.005) had a significant effect on purchase intention, respectively. However, the attitude towards sustainability for the single notion had a significant effect on PI (β = 0.032, *p* < 0.1). Therefore, we can infer that when consumers’ perception of sustainability affects their purchase intention of the sustainable product, consumers tend to portray sustainability as a whole. Therefore, we can infer that the impact of ATS on PI was significant in the holistic perception of sustainability; therefore, H1 is partially supported, and the results are shown in Table 3.

The moderation effect of sustainability-type STOs on the impact of ATS on PI.

We then investigated how customers perceive sustainability overall when only attitude influences the purchase intention. When it comes to a trade-off with sustainability, consumers may have different reactions based on the perceived value of different sustainability attributes [111,112]; therefore, an examination was conducted to assess the moderating effect of sustainability-type trade-offs (i.e., social vs. environmental) on the impact of the ATS on PI. Specifically, a study was developed comprising four different scenarios through the classification of sustainability into two main categories—environmental and social sustainability. The aim was to investigate how the moderating impacts of the sustainability category and its strength affect customers’ STO (sustainability trade-off) choices. Nevertheless, the findings did not demonstrate noteworthy results for the interplay among sustainability category, strength, and compromises. A logistic regression analysis was executed utilizing Model 1 of the PROCESS macro introduced by Hayes (2018) [113], employing bootstrapping with 5000 samples and a 95% confidence interval.

The indirect effect of ATS on PI through STOs in sustainability type was not significant, that is, the interaction effect of ATS on STOs with environmental sustainability (β = −0.18, *SE* = 0.17, *p* > 0.05, 95% CI [−0.50, 0.13]) and on STOs with social sustainability (β = 0.18, *SE* = 0.16, *p* > 0.05, 95% CI [−0.13, 0.50]). This suggests that the purchase intention for sustainable products is led by the attitude towards sustainability and does not suffer any interference from any types of sustainability trade-offs. This attitude towards sustainability suggests that the different types of STOs had no moderation effect. Therefore, H2 is not supported, and the results are shown in Table 4.

The moderation effect of attribute-type STOs on the impact of ATS on PI.

When consumers face a sustainability trade-off, they not only face sustainability-type STOs but also a trade-off with the product attribute. Since utilitarian attributes are often seen as essential and non-negotiable, whereas hedonic attributes are perceived as additional benefits that can be sacrificed for a greater good, different outcomes may result when STOs with product attributes intervene in the relationship between attitude towards sustainability and purchase intention. Therefore, we conducted an analysis using PROCESS Macro Model 1 to explore the mediating effect of attribute-type STOs on the relationship between ATS and PI. In a scenario, we set the STO with utilitarian and hedonic attribute values, as shown in Figure 2. Therefore, the intensities of the attribute trade-off with sustainability, that is, performance (design) (9 or 5), were transformed into quantitative variables through dummy coding. Since we had four variables representing the attribute value (high vs. low) and type of attribute (utilitarian vs. hedonic) in STOs, four dummy variables were used. However, the indirect effect of ATS on PI through attribute-type STOs was not significant in all variables: attribute type 1 (high utilitarian) b = −0.19, SE = 0.19, *p* > 0.05, 95% CI [−0.55, 0.18]; attribute type 2 (low utilitarian) b = −0.05, SE = 0.18, *p* > 0.05, 95% CI [−0.42, 0.31]; attribute type 3 (high hedonic) b = 0.05, SE = 0.19, *p* > 0.05, 95% CI [−0.31, 0.42]; attribute type 4 (low hedonic) b= −0.19, SE = 0.19, *p* > 0.05, 95% CI [−0.55, 0.18]; this suggests that neither sustainability- nor attribute-type STOs significantly affected the impact of ATS on PI. Therefore, H3 is not supported, and the results are shown in Table 5.

The interaction of sustainability- and attribute-type STOs in relation to the impact of ATS on PI.

When consumers have STOs in purchasing, contextual factors may affect in the purchase [8,9,12,13,25]; we further investigate the interactions of STOs with PI excluding the personal factors as their ATS. We ran an ad hoc analysis on the interaction between attribute- and sustainability-type STOs in relation to PI. The analysis performed with PROCESS Macro Model 3 focused on the moderating effect of attribute- and sustainability-type STOs on the ATS–PI relationship.

We initially explored the interaction effect of ATS on STOs within each division regarding PI. However, the interaction between ATS and sustainability- and attribute-type STOs does not demonstrate a significant impact on PI. Hence, sustainability- and attribute-type STOs are solely insufficient factors for valid interactions with ATS. This suggests that these factors alone are insufficient to guarantee a specific outcome in explaining the attitude–behavior gap in the sustainable product market. 

The results of the three-way interaction action representing the interaction effect of ATS with STOs on environmental sustainability did not reach statistical significance (*p* > 0.05), suggesting that the joint influence of ATS and environmental sustainability trade-offs (STOEs) does not have a substantial impact on purchase intention (PI). Similarly, the interaction effect of ATS with STOs on utilitarian product attributes also did not show statistical significance (*p* > 0.05), indicating that the combination of ATS and high utilitarian attributes (STOU_high_) does not notably influence PI. In contrast, the three-way interaction involving ATS*STOs on environmental sustainability*STOs on utilitarian attributes is statistically significant (*p* < 0.05), revealing a significant negative effect on PI when considering ATS, environmental sustainability trade-offs (STOEs), and high utilitarian attributes (STOU_high_) collectively. Conversely, the three-way interaction effect with ATS*STOs on environmental sustainability*STOs on hedonic attributes is not statistically significant (*p* > 0.05). The significant negative interaction effect among ATS, STOEs, and STOU_high_ indicates a noticeable decrease in the purchase intention when all three factors are taken into account. This underscores a complex interplay among consumer attitudes towards sustainability, environmental trade-offs, and high utilitarian attributes, resulting in a reduced likelihood of purchase. The lack of significant interactions implies that the individual or pairwise effects of these variables do not substantially impact the purchase intention, underscoring the importance of examining the combined influences of multiple factors when evaluating consumer behavior in relation to sustainability. The results are shown in Table 6.

Ad hoc *analysis*: The Effect of Sustainability- and Attribute-Type STOs on PI.

To revalidate whether purchase intentions significantly differ depending on the type of sustainability and the specific trade-offs associated with product characteristics, an independent sample *t*-test was conducted. The results indicated significant differences in purchase intentions based on the trade-off between practical versus hedonic qualities in the context of environmental sustainability (*t* = 3.840, *p* < 0.01). In contrast, for social sustainability, no significant differences were detected in purchase intentions based on the characteristics contrasted. Furthermore, when trading off functional value, consumers showed more pronounced differences in the context of environmental sustainability. We have shown results in Figure 3. In Figure 3, a comparative result is presented regarding the impacts of sustainability trade-offs (STO) on purchase intention across various sustainability and attribute types. The placement of the two charts side by side within the figure is deliberate, aiming to emphasize how the influence of STO varies based on the specific sustainability attributes being exchanged. Figure 3 demonstrates that the impact of STO on purchase intention is not consistent but rather varies according to different sustainability and attribute types. 

The mediation effect of moral regulation on the impact of ATS on PI.

An examination was conducted to assess the mediation effect of moral licensing in relation to the impact of ATS on PI in an STO situation. We conducted a mediation analysis using the PROCESS macro for SPSS ([113], Model 4) to examine the role of *Moral Regulation* in the relationship between ATS (attitude towards sustainability) and purchase intention. The indirect effect of ATS on PI through moral regulation was tested using a bootstrap estimation approach with 5000 bootstrap samples. The direct path from ATS to MR was significant: β = 0.36, SE = 0.04, *t* = 8.39, *p* < 0.01, 95% CI [0.2783, 0.4485]; this suggests a strong positive relationship between ATS and MR. Similarly, the path from MR to PI was also significant: β = 0.47, SE = 0.10, *t* = 4.61, *p* < 0.01, *p* < 0.01, 95% CI [0.2674, 0.6637]; this indicates a significant positive effect of MR on PI. Lastly, the direct effect of ATS on PI was statistically significant: β = 0.20, SE = 0.09, *t* = 2.02, *p* < 0.05, and 95% CI [0.0058, 0.3925]. We also investigated the mediation effect of ‘morality’ on the impact of ATS on *p* for the control condition to clarify the moderation effect of moral licensing. The direct path from ATS to morality was significant: β = 0.44, SE = 0.04, *t* = 11.50, *p* < 0.01, and 95% CI [0.3628, 0.5137]; this suggests a strong positive relationship between ATS and morality. However, the path from morality to PI was not significant: β = 0.10, SE = 0.10, *t* = 0.94, *p* > 0.05, and 95% CI [−0.1103, 0.3131]. Lastly, the direct effect of ATS on PI was statistically significant: β = 0.31, SE = 0.10, *t* = 2.98, *p* < 0.01, and 95% CI [0.1060, 0.5116]. Therefore, the results clarify that not the morality but the moral regulation of a consumer affects the context of STOs in sustainable product purchasing.

## 5. Discussion

This investigation concentrated on various factors that impact consumers’ intentions to buy sustainable products, specifically emphasizing sustainability trade-offs (STOs) and moral regulation. The findings illustrated numerous critical insights into the interaction and influence of these factors on consumer behavior. Consider H1: the attitudes towards environmental and social sustainability on purchasing sustainable products will be examined. However, the analysis indicated that the overall attitudes towards sustainability (ATS_t_) had a significant influence on purchase intention (PI), whereas attitudes towards environmental (ATS_E_) and social sustainability (ATS_S_) individually did not have a noteworthy effect. This suggests that consumers tend to perceive sustainability holistically rather than categorizing it into distinct dimensions. The analysis for H2 indicates that neither trade-offs in environmental (STOE) nor social sustainability (STOS) significantly moderated the impact of ATS on PI. This implies that trade-offs between different types of sustainability do not affect the impact of overall sustainability attitudes on purchasing decisions. In addition, we investigated the moderation effect of attribute-type STOs on the relationship between ATS and PI. The results show that the moderation effect of sustainability trade-offs with utilitarian (STOU) and hedonic (STOH) attributes was also not significant. None of the interactions, whether involving high or low levels of utilitarian or hedonic attributes, exhibited a significant influence (all *p*-values > 0.05). This finding implies that trade-offs involving product attributes do not independently moderate the connection between sustainability attitudes and purchase intentions. Through the ad hoc analysis, the combined moderation effect of sustainability- and attribute-type STOs on the relationship between ATS and PI was investigated. Specifically, the three-way interaction (ATS*STOE*STOU_high_) was significant (β = −1.58, *p* < 0.05), indicating that the combined impact of these trade-offs significantly affects purchase intentions. This implies that consumers are particularly responsive to trade-offs involving environmental benefits and practical product features. Proposition 5: The mediating role of moral regulation in the relationship between environmental concern and purchase intention in STO conditions will be explored. Table 7 illustrates that moral regulation (MR) acts as a mediator between ATS and PI. The paths from ATS to MR (β = 0.36, *p* < 0.01) and from MR to PI (β = 0.47, *p* < 0.01) were significant, as was the direct path from ATS to PI (β = 0.20, *p* < 0.05). This suggests that consumers’ purchase decisions are significantly influenced by their moral regulation processes, aligning their sustainability attitudes with their purchase behaviors.

## 6. Conclusions

This study explores the complex interrelations between consumers’ viewpoints regarding sustainability and their buying behaviors, concentrating on sustainability trade-offs (STOs) and ethical oversight. This investigation presents several fundamental insights that enrich our comprehension of eco-friendly consumption dynamics.

Initially, our discoveries unveiled that consumers’ comprehensive stance towards sustainability (ATSt) significantly affects their desire to acquire sustainable goods. Interestingly, upon investigating attitudes towards environmental (ATS_E_) and societal sustainability (ATS_S_) separately, they did not exhibit a remarkable influence. This indicates that consumers tend to perceive sustainability as an all-encompassing concept rather than discerning between its ecological and societal facets. This comprehensive outlook suggests that marketing approaches should highlight the general advantages of sustainability instead of singling out specific elements.

Secondly, the analysis demonstrated that sustainability both in environmental (STOE) and societal sustainability (STOS) did not notably moderate the association between consumer’s attitude towards sustainability and the purchase intentions (PI). When consumers engage in the STO, types of sustainability do not distinctively modify the impact of overall sustainability attitudes on purchasing choices. Similarly, for the product side, the moderating impact of sustainability compromises with utilitarian (STOU) and hedonic characteristics (STOH) on the link between ATS and PI was also insignificant. However, when the two types of STO combines, they significantly exert an impact on the relationship of ATS to PI. This shows that in STO, both sustainability type and product attributes are important factors affecting the purchase intention of the sustainable product. Even more, when we conducted the ad hoc analysis, it showed that beside participants ATS, the interaction between two kinds of STO still impacts the purchase intention. To apply this notion in solving the attitude–behavior gap held in sustainability areas, we can reconfirm that the contextual factor matters. This revalidates previous research [12,31,38,39] positing that contextual factors such as functional and symbolic product attributes, green as core value for product, etc., affect the purchase intention. Nevertheless, our ad hoc analysis uncovered a significant three-way interaction (ATS*STO_E_*STOU_high_), underscoring that the collective influence of these compromises substantially impacts purchase intentions. This indicates that consumers are particularly responsive to compromises involving environmental advantages and practical product attributes. This nuanced discovery emphasizes the significance of considering combined compromises in marketing sustainable goods as consumers react more favorably when practical attributes are coupled with environmental advantages.

In addition, the inquiry examined the mediating role of moral regulation in the association between attitudes towards sustainability and intentions to buy. In contrast to previous studies that suggested that the level of ethics or morality in consumers influences sustainable consumption [3,18,24,105], this study confirms that the relative morality of consumers has a greater impact on their intentions to purchase sustainable products. This implies that consumers’ past actions based on their morals have a stronger influence on their ethical conduct, including the purchase of sustainable products. This alignment of sustainability attitudes with buying behaviors through ethical oversight processes like moral licensing and cleansing underscores the psychological intricacy inherent in sustainable consumption. Based on the findings of this study, it can be argued that the ‘conditional morality’ of consumers may have a more significant impact on their decisions to purchase sustainable products.

In summary, this inquiry provides numerous vital insights into the determinants of sustainable product acquisitions. The outcomes suggest that consumers perceive sustainability as a holistic concept, substantially impacting their purchase intentions. While individual viewpoints towards environmental or social sustainability do not independently affect purchase decisions, the overall perception of sustainability carries significant weight. The study also sheds light on the complex nature of sustainability trade-offs. Although individual compromises among different sustainability aspects and product characteristics do not substantially alter the relationship between sustainability attitudes and purchase intentions, the combined interaction of environmental sustainability and utilitarian attributes does indeed have a notable impact. Furthermore, the pivotal role of moral regulation in sustainable consumption implies that marketing strategies should focus on aligning product attributes with consumers’ ethical oversight processes.

## 7. Implications and Limitations

This article puts forward a new outlook that contrasts with Ajzen’s Theory of Planned Behavior (TPB) and Stern’s Norm Activation Theory, which propose that personal morality plays a role in buying sustainable products. However, this study suggests that contextual factors and situational mechanisms have a greater impact on sustainable product purchases. It presents a framework combining sustainability trade-offs and attitudes towards sustainability to understand consumer motivations. The study explores how social and environmental sustainability and utilitarian and hedonic features affect consumer decision-making in sustainable buying choices. The study’s detailed perspective on sustainability trade-offs enhances the comprehension of how various factors influence sustainable product purchases and bridges the gap between attitudes and behaviors. Identifying moral regulation as an intensifying factor between sustainability attitudes and purchase intent is a significant theoretical contribution. Emphasizing the importance of moral regulation, the research adds a crucial element to existing consumer behavior theories, showing that sustainable consumption is influenced by more than just attitudes and perceived benefits.

The discoveries within this research offer numerous crucial implications for managers looking to boost sustainable product purchase intentions. Initially, it is essential for managers to recognize that consumers often see sustainability as a vague and abstract concept. To bridge this perceptual divide, marketing strategies should focus on clearly explaining the concrete benefits of sustainable products. Moreover, the study emphasizes the importance of strategically showcasing product attributes in the marketing of sustainable products. Consumers respond differently to sustainability trade-offs depending on whether the product attribute is utilitarian or hedonic. Managers should prioritize highlighting the practical benefits of environmental sustainability, like tangible advantages and long-term savings. When communicating sustainability benefits, managers can tap into consumers’ moral regulation by proposing varying levels of moral attributes in the product or marketing messages to encourage consumers to choose sustainable products due to moral considerations.

Several limitations were identified in this research. This study only focused on sustainability trade-offs within the product sector, overlooking social and environmental interactions in service contexts. Further research is needed to understand the link between sustainability types and purchase intentions in service industries. In addition, in the methodology used, self-reported measures could introduce bias, while utilizing more objective approaches or longitudinal studies might offer more reliable data. The complexity of sustainability trade-offs may not have been fully captured. Other dimensions and interactions were not considered in this study. Scrutinizing these elements and utilizing unbiased measures can contribute to a more thorough understanding of the association between sustainability attitudes and purchase intentions.

## Figures and Tables

**Figure 1 behavsci-14-00702-f001:**
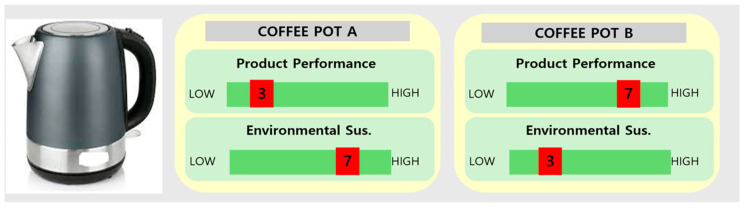
Examples of sustainability trade-offs (STOs).

**Figure 2 behavsci-14-00702-f002:**
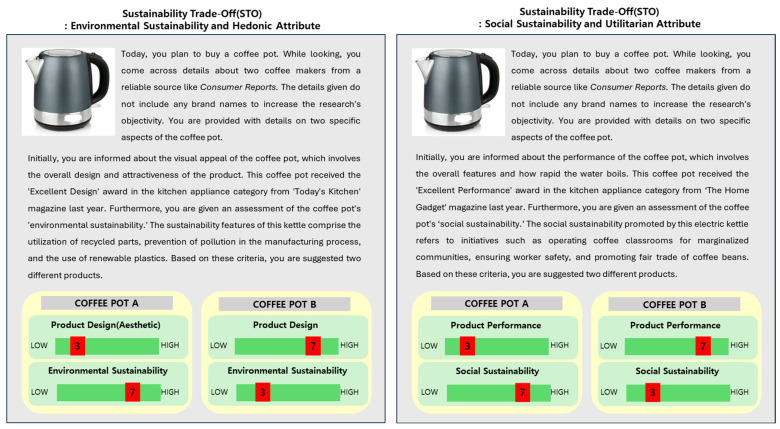
Examples of sustainability trade-off scenarios.

**Figure 3 behavsci-14-00702-f003:**
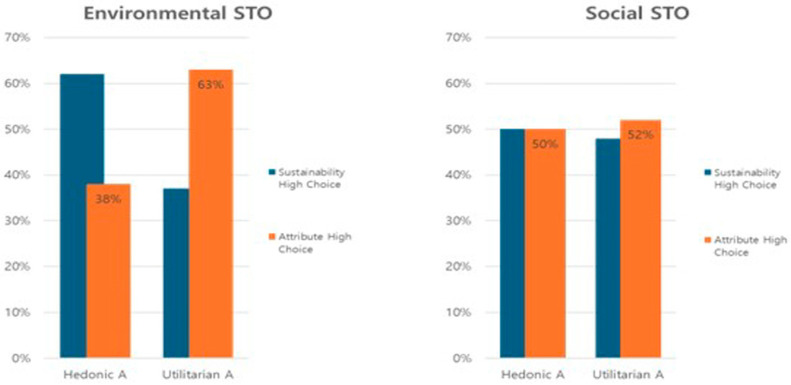
The comparison of the effect of sustainability- and attribute-type STOs on PI.

**Table 1 behavsci-14-00702-t001:** Demographic of the participants.

Domain	Details	Freq.	Percentage
Gender	Male	238	52
	Female	219	48
	Total	457	100
Age	19–29	162	35.5
	30–39	93	20.4
	40–49	128	28
	Over 50	105	16.1
	Total	457	100
Profession	Student	145	31.6
	Housewife	28	6.1
	Fulltime Employment	179	39
	Other	105	22.9
	Total	457	100

**Table 2 behavsci-14-00702-t002:** Factor analysis and reliability analysis of the independent variables.

Variables	Items	Loadings	Cronbach’s Alpha
Attitude to Sustainability	*Issues of social sustainability are important to me*	0.892	0.842
	*Environmental sustainability issues are important to me*	0.834	
	*It is important to me that companies hold high ethical standards*	0.774	
Moral Regulation	*The action was consistent with my moral principles*	0.875	0.798
	*After the choice, I feel I have exerted adequate effort to fulfill my moral objective*	0.891	
Morality	*I try to handle most things honestly*	0.710	0.710
	*I always act in a way that maximizes benefits to others and minimizes harm*	0.805	
	*Not hurting others is one of the principles of my life*	0.770	
	*KMO = 0.783*, *Bartlette’s x² = 1354.638 (p < 0.001)*		

**Table 3 behavsci-14-00702-t003:** The effect of different types of attitudes towards sustainability to purchase intention.

DV	IV	b	S.E.	β	*t*	*p*	VIF
Purchase Intention	(constant)	0.063	0.110		0.572	0.567	
	ATS_E_	0.076	0.029	0.032	0.648	0.648	2.242
	ATS_S_	0.032	0.025	0.076	0.258	0.258	2.135
	ATS_t_	0.032	0.025	0.104	0.095	0.095	1.806
	F = 5.424 (*p* < 0.001). R^2^ = 0.035 _adj_R^2^ = 0.028, D-W = 1.943						

Notes. ATS (attitude towards sustainability), ATS_E_ (attitude towards environmental sustainability), ATS_S_ (attitude towards social sustainability), ATS_T_ (attitude towards total sustainability). DV: Dependent Variable, IV: Independent Variable, VIF: Variance Inflation Factor.

**Table 4 behavsci-14-00702-t004:** The effect of sustainability-type STOs on the impact of ATS on PI.

**PI**	**Interaction**		**β**	**S.E.**	** *p* **	**CI**
	ATS*	STOE	−0.18	0.17	>0.05	[−0.50, 0.13]
		STOS	0.18	0.16	>0.05	[−0.13, 0.50]

Notes. PI (purchase intention), ATS (attitude towards sustainability), STOE (sustainability trade-offs in environmental sustainability), STOS (sustainability trade-offs in social sustainability), ATS*STOE (ATS interaction with STOE), ATS*STOS (ATS interaction with STOS).

**Table 5 behavsci-14-00702-t005:** The effect of attribute-type STOs on the impact of ATS on PI.

**PI**	**Interaction**		**β**	**S.E.**	** *p* **	**CI**
	ATS*	STOU_high_	−0.19	0.19	>0.05	[−0.43, 0.24]
		STOU_low_	−0.05	0.18	>0.05	[−0.35, 0.29]
		STOH_high_	0.05	0.19	>0.05	[−0.29, 0.34]
		STOH_low_	−0.19	0.19	>0.05	[−0.29, 0.34]

Notes: PI (purchase intention), ATS (attitude towards sustainability), STOU_high_ (sustainability trade-offs with a high utilitarian attribute value), STOU_low_ (sustainability trade-offs with a low utilitarian attribute value), STOH_high_ (sustainability trade-offs with a high hedonic attribute value), and STOH_low_ (sustainability trade-offs with a low hedonic attribute value).

**Table 6 behavsci-14-00702-t006:** The interaction effect of sustainability- and attribute-type STOs on the impact of ATS on PI.

**PI**	**Interaction**	**β**	**S.E.**	*p*	**CI**
	ATS*STOEs	−0.03	0.06	>0.05	[−0.14, 0.07]
	ATS*STOU_high_	−0.41	0.40	>0.05	[−1.19, 0.37]
	**ATS*STOEs*STOU_high_**	**−1.58**	**0.76**	**<0.05**	**[−3.08, −0.08]**
	ATS*STOEs*STOH_high_	0.18	0.14	>0.05	[−0.07, 0.45]

Notes: PI (purchase intention), ATS (attitude towards sustainability), STOE (environmental sustainability trade-offs), STOU_high_ (sustainability trade-offs with high utilitarian attribute value), STOU_low_ (sustainability trade-offs with low utilitarian attribute value), and STOH_high_ (sustainability trade-offs with high hedonic attribute value).

**Table 7 behavsci-14-00702-t007:** The mediation effect of moral regulation on the impact of ATS on PI.

Path	β	S.E.	*t*	*p*	CI
ATS to MR	0.36	0.04	8.39	<0.01	[0.2783, 0.4485]
MR to PI	0.47	0.10	4.61	<0.01	[0.2674, 0.6637]
ATS to PI	0.20	0.09	2.02	<0.05	[0.0058, 0.3925]

Note. ATS (attitude towards sustainability), PI (purchase intention), MR (moral regulation).

## Data Availability

The data used in the research were gathered in two parts. First, data were collected from 159 students who attended courses run by the authors in Pusan National University and Ulsan College. Second, data were collected from 260 participants through an online research institution. For inquiries regarding the data requirements, please contact the corresponding author (jeyoo@ks.ac.kr).

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
