# Peer review of "Exploring the Impact of Sustainability Trade-Offs: The Role of Product and Sustainability Types in Consumer Purchases Mediated by Moral Regulation"

_behavsci, 2024, doi:10.3390/bs14080702_

Round 1
Reviewer 1 Report
Comments and Suggestions for Authors
This study examines contextual factors in sustainable consumption, particularly sustainability trade-offs (STO) faced by consumers during product purchase and the impact of conditional morality. The STO is analyzed in terms of sustainability type (social vs environmental) and attribute type (utilitarian vs hedonic).
Sample: 457 participants in South Korea
The article is well written and uses several bibliographic references, but some are already a few years old. But the article must be improved.
The manuscript must be improvement in some issues:
- Cannot use the same acronym for two different things. (e.g.: sustainability trade-off (STO)and strategic trade-off (STO) -line 96)
- Verify the references (e.g.: line 137, 145, 162, 208…)
- The H2. STO in Sustainability Type will moderate the effect on ATS to PI. (line 218), but the introduction of the acronym is only in the line 380
- Normally the structure of the literature review, for each point will be the formulation of a hypothesis. (e.g.: Shouldn't point 2.2 include be before H1? i.e. 2.1 and 2.2 be just one point?).
- Explain better the lines 320-323. If the questionnaire was conducted on 459 South Korean participants from Busan National University and Ulsan College, why table 1 have Housewife?
- Research questions must be coordinated with hypotheses. The results must answer the research questions and validate or not the hypotheses. The article must be improved so that this coordination is clear.
Author Response
|
Comments 1: This study examines contextual factors in sustainable consumption, particularly sustainability trade-offs (STO) faced by consumers during product purchase and the impact of conditional morality. The STO is analyzed in terms of sustainability type (social vs environmental) and attribute type (utilitarian vs hedonic). Sample: 457 participants in South Korea The article is well written and uses several bibliographic references, but some are already a few years old. But the article must be improved. |
|
Response 1: We appreciate your feedback on the bibliographic references. While our article includes several references, we acknowledge that some are a few years old. We reviewed and updated our references to include more recent studies, ensuring our work is informed by the latest research and developments. This will improve the article's relevance and strengthen its contribution to the field.
|
|
Comments 2: Cannot use the same acronym for two different things. (e.g.: sustainability trade-off (STO)and strategic trade-off (STO) -line 96). |
|
|
|
Response 2: Thank you for highlighting the oversight regarding the use of the acronym "STO" for both "sustainability trade-off" and "strategic trade-off." To eliminate any ambiguity and ensure clarity in our manuscript, we have revised the acronyms STO for one acronym for ‘sustainability trade-offs’.
Comments 3: Verify the references (e.g.: line 137, 145, 162, 208…)
Response 3: Thank you for your comment. We apologize for the oversight regarding the missing references. We have thoroughly reviewed the manuscript and identified the missing references at the indicated lines (137, 145, 162, 208, etc.). We have now added the appropriate references to these sections to ensure completeness and accuracy. Please find the updated manuscript with the corrected references attached. We appreciate your attention to detail and thank you for helping us improve the quality of our work. We verify the references as followed:
Line 138: [75]WCED, S.W.S. World commission on environment and development. Our common future 1987, 17, 1-91. Line 145: [76]Gupta, A.; Dash, S.; Mishra, A. All that glitters is not green: Creating trustworthy ecofriendly 9services at green hotels. Tourism Management 2019, 70, 155-169. Line 168 [5]: Cowe and William(2000) was wrongly written. I am very sorry. We modified into Cowe and Simon : Crowe, R.; Simon, W. Who are the ethical consumers?; Citeseer: 2000. Line 217 [5]: For the Bruntland Report we verify the references as Brundtland, G. H. (1987). What is sustainable development. Our common future, 8(9).
Comments 4: The H2. STO in Sustainability Type will moderate the effect on ATS to PI. (line 218), but the introduction of the acronym is only in the line 380
Response 4: Thank you for pointing out the delayed introduction of the acronym "STO" in our manuscript. We realize the importance of presenting acronyms clearly and at their first point of use to ensure readability and coherence in the narrative. We added the explanation of ATS on line 158 and 166. in H1 and we have mentioned the purchase intention in line 186.
Comments 5: Normally the structure of the literature review, for each point will be the formulation of a hypothesis. (e.g.: Shouldn't point 2.2 include be before H1? i.e. 2.1 and 2.2 be just one point?)
Response 5: Thank you for your insightful comment regarding the organization of our literature review and the placement of hypotheses. We appreciate your suggestion to streamline the presentation by integrating points 2.1 and 2.2, ensuring a logical progression that naturally leads into Hypothesis 1 (H1). Upon reviewing the sections in question, we agree that combining these points would enhance the flow of the argument and provide a more cohesive foundation for the hypotheses. Consequently, we have revised the literature review to merge the discussions in points 2.1 and 2.2. This modification allows us to more clearly establish the theoretical groundwork before introducing H1, thereby improving the logical sequence and readability of our manuscript. We merged as follow:
“Attitudes towards sustainability are often shaped by the need to navigate and manage trade-offs between various sustainability dimensions, such as economic, environmental, and social aspects. The concept of attitude towards sustainability (ATS) pertains to the level of importance consumers place on sustainability and how they prioritize it when making purchases [40]. This mindset encompasses various considerations, including the social, environmental, and ethical aspects of sustainability. ATS plays a significant role in how consumers weigh the trade-offs between sustainability and other attributes of a product, such as hedonic (e.g., esthetics) and utilitarian (e.g., functional performance) values, impacting their emotional responses and decision-making processes. Environmental concerns are important in the shift towards sustainable goods, but a gap between consumer attitudes and actions towards sustainability remains[85]. Cowe and Williams (2000) highlighted that even though 30 percent of consumers demonstrated an inclination to buy eco-friendly items, these purchases represented less than 3 percent of the overall market share [5]. This disparity is widely recognized in the academic literature and is commonly known as ‘the green attitude–behavior gap’ [8], ‘the green intention–behavior gap’ [26], or ‘the motivation–behavior gap’ [85]. Currently, numerous studies are dedicated to elucidating, comprehending, and addressing this phenomenon [7] [12] [11] [27]. Elhaffar et al. [25] have argued that research focused on the motivations behind and barriers to sustainable consumption broadly lack focus on the green gap phenomenon and the methodologies used in addressing it. Most academic research utilizes the economic rational paradigm to examine the green gap phenomenon primarily through the Theory of Planned Behavior (TPB), which enforces a specific alignment between attitudes, subjective norms, and perceived control. In contrast to conventional purchase behavior, where the binary options of ‘purchase’ or ‘non-purchase’ exist, in sustainable purchase behavior, the consumer categories become more varied, including ‘green gapper’ between ‘non-green consumer’ and ‘green consumer’ [12]. Though consumer attitudes, norms, and other intrapsychic factors are the not the only determinants of sustainable behavior [34,86], contextual elements serve as significant variables impacting the sustainability gap, influencing it either in a positive or negative manner exerting to consumer’s purchase intention(PI) on sustainable products. Not only do one’s personal norms and altruism affect PIs for sustainable products; other contextual factors also impact on PI for the sustainable product. That is purchasing of sustainable products often necessitates trade-offs for users, such as increased prices, diminished quality, or reduced performance, resulting in an attitude–behavior gap. Considering the trade-offs, the scope of sustainability in social, environmental, and economic dimensions [32,86,87] can be applied as an attribute. Consumers undergo a psychological process of trading off sustainability within the product against other attributes in evaluation [86] [39]. Regard to previous research [40], this study defines the term sustainability trade-off (STO) as ‘the exchange in the ratio of the product attribute (A) that planned to provide the intrinsic value to the consumer with the attribute of sustainability (B)’. The approach does not entail selecting one characteristic over the other but rather the extent to which they are factored in with a consistent total value of 10, indicating that the overall value stays constant regardless of the changing proportion between sustainability and product features. Examples of STOs are shown in Figure 1.”
Comments 6: Explain better the lines 320-323. If the questionnaire was conducted on 459 South Korean participants from Busan National University and Ulsan College, why table 1 have Housewife
Response 6: Thank you for your astute observation regarding the participant demographics reported in lines 320-323 and the data presented in Table 1. We acknowledge the confusion caused by the inclusion of the category "Housewife" in the table, given the primary participant group comprised students from Busan National University and Ulsan College. We had two participant groups when gathering data, we didn’t mention it in detail. Thank you for the precise notice. We put further explanation for the participants and its demographic as follows: An open-ended questionnaire was conducted on 199 South Korean participants from Busan National University and Ulsan College. A follow-up survey was done on 260 participants through an online research institution(http://surveybilly.com). 457 responses out of 459 were used for the analysis, the details of the participants are shown in Table 1.
Comments 7: Research questions must be coordinated with hypotheses. The results must answer the research questions and validate or not the hypotheses. The article must be improved so that this coordination is clear.
Response 7: Thank you for your constructive feedback regarding the alignment of our research questions, hypotheses, and the presentation of results. We recognize the importance of ensuring that these elements are clearly connected throughout the manuscript to provide a coherent narrative and substantiate our findings. We modified the RQ as follows.
RQ1. How does the consumer's attitude towards sustainability influence their tendency to purchase sustainable products? RQ2. In the realm of sustainable product purchasing, does the contextual factors besides the morality of consumer impact STOs (sustainability trade-offs)? Specifically, do the types of sustainability and the attributes of the product hold substantial importance within the STO framework?? RQ3. What are the underlying psychological mechanisms in STOs and how do they impact the consumer's decision to purchase sustainable products?
Also, we put the result for the research questions in conclusion part. As follows:
“This article puts forward a new outlook that contrasts with Ajzen's Theory of Planned Behavior (TPB) and Stern's Norm Activation Theory that propose the personal morality plays a role in buying sustainable products. However, the study suggests contextual factors and situational mechanisms have a greater impact on sustainable product purchases. It presents a framework combining sustainability trade-offs and attitudes towards sustainability to understand consumer motivations. The research explores how social and environmental sustainability, utilitarian, and hedonic features affect consumer decision-making in sustainable buying choices. The study's detailed perspective on sustainability trade-offs enhances comprehension of how various factors influence sustainable product purchases and bridge the gap between attitudes and behaviors. Identifying moral regulation as an intensifying factor between sustainability attitudes and purchase intent is a significant theoretical contribution. Emphasizing the importance of moral regulation, the research adds a crucial element to existing consumer behavior theories, showing sustainable consumption is influenced by more than just attitudes and perceived benefits. The discoveries of this research offer numerous crucial implications for managers looking to boost sustainable product purchase intentions. Initially, it is essential for managers to recognize that consumers often see sustainability as a vague and abstract concept. To bridge this perceptual divide, marketing strategies should focus on clearly explaining the concrete benefits of sustainable products. Moreover, the study emphasizes the importance of strategically showcasing product attributes in the marketing of sustainable products. Consumers respond differently to sustainability trade-offs depending on whether the product attribute is utilitarian or hedonic. Managers should prioritize highlighting the practical benefits of environmental sustainability, like tangible advantages and long-term savings. When communicating sustainability benefits, managers can tap into consumers' moral regulation by proposing varying levels of moral attributes in the product or marketing messages to encourage consumers to choose sustainable products due to moral considerations. Several limitations were identified in the research. The study only focused on sustainability trade-offs within the products sector, overlooking social and environmental interactions in service contexts. Further research is needed to understand the link between sustainability types and purchase intentions in service industries. Also in methodology, self-reported measures could introduce bias, while utilizing more objective approaches or longitudinal studies might offer more reliable data. The complexity of sustainability trade-offs may not have been fully captured. Other dimensions and interactions were not considered in the study. Scrutinizing these elements and utilizing unbiased measures can contribute to a more thorough understanding of the association between sustainability attitudes and purchase intentions.”
|

Reviewer 2 Report
Comments and Suggestions for Authors
This is very interesting research conducted in an interesting way, supported by in-depth literature research. As mentioned in the article, this research may have serious practical implications for marketing and managers' approach to presenting their products.
Author Response
|
3. Point-by-point response to Comments and Suggestions for Authors Nonapplicable
|
|
4. Response to Comments on the Quality of English Language |
|
Nonapplicable
|
|
|
|
|
|
5. Additional clarifications
- Final Appreciation to the Reviewer
Thank you very much for your insightful feedback and valuable comments on our manuscript. Your suggestions have been instrumental in enhancing the clarity and quality of our work. We are grateful for your time and effort in reviewing our paper, and we have carefully addressed all your comments in the revised version. We appreciate your support and hope that the revisions meet your expectations.
Sincerely,
Je Eun Yoo Corresponding Author.
|

Reviewer 3 Report
Comments and Suggestions for Authors
This research investigates sustainability trade-offs in terms of sustainability type and attribute type and indicates that moral regulation enhances the impact of sustainable trade-offs on purchase intention. They conducted four experiments in order to conclude their empirical results. This research shows the moderating role of sustainable trade-offs in the relationship between attitudes towards sustainability and purchase intention, underscoring the importance of contextual factors in sustainable product purchase. Firms can leverage sustainable trade-offs in their marketing strategies, incorporating product features and advertising messages. The topic is somewaht noble. I don't find any technical errors. However, the current manuscript has many issues to improve in order to be published in this journal as follows:
1. ATS did not defined. If you wish, you can use abbreviation, however, you carefully defined it for the first time that you used in the manuscript.
2. 'moral regulation' is not clearly defined in the section 2.4. Hence H5 is not easy to follow.
3. In line 391, you would be better to use 'significant' instead of 'valid'.
4. In table 4, three way interaction should be carefully interpreted.
5." Author Contributions: “Conceptualization; methodology; software; validation; formal analysis; 578 investigation; resources; data curation; writing—original draft preparation; writing—review and 579 editing, Je Eun Yoo." sounds the co-author is free-rider. Please also recognize the contribution of the co-author. If the current description is true then it should be single author paper.
Good luck.
Comments on the Quality of English Language
Minor copy-edit would be nice.
Author Response
|
Comments 1: This research investigates sustainability trade-offs in terms of sustainability type and attribute type and indicates that moral regulation enhances the impact of sustainable trade-offs on purchase intention. They conducted four experiments to conclude their empirical results. This research shows the moderating role of sustainable trade-offs in the relationship between attitudes towards sustainability and purchase intention, underscoring the importance of contextual factors in sustainable product purchase. Firms can leverage sustainable trade-offs in their marketing strategies, incorporating product features and advertising messages. The topic is somewhat noble. I don't find any technical errors. However, the current manuscript has many issues to improve in order to be published in this journal as follows:
|
|
Response 1: We sincerely appreciate the valuable feedback provided by the referee on our paper. The comments have helped us improve the quality and clarity of our study. We have carefully considered the reviewer's comments and have made significant revisions to our paper to address the reviewer’s concerns. We have carefully revised our paper, incorporating the insightful comments provided by the reviewer. These suggestions have significantly enhanced the quality and rigor of our study. Our revised paper effectively addresses the concerns raised by the reviewer and contributes to the ongoing discussion on the impact of sustainability trade-offs in the marketing region.
|
|
Comments 2: ATS did not defined. If you wish, you can use abbreviation, however, you carefully defined it for the first time that you used in the manuscript. |
|
|
|
Response 2: Thank you for pointing this out. With considering all the excellent comments from the reviewer, we have revised nearly all parts of the paper from the introduction to the conclusion. While considering all the comments from reviewer, the author was able to add the definition of Attitude Towards Sustainability and noted at introduction in line 118 and section 2.1 ‘Consumer’ Attitude toward Sustainability’. We have put more explanation and elaboration on the attitude toward sustainability in the section 2.1 line 158 to 166 as follows. “Attitudes towards sustainability are often shaped by the need to navigate and manage trade-offs between various sustainability dimensions, such as economic, environmental, and social aspects. The concept of attitude towards sustainability (ATS) pertains to the level of importance consumers place on sustainability and how they prioritize it when making purchases [40]. This mindset encompasses various considerations, including the social, environmental, and ethical aspects of sustainability. ATS plays a significant role in how consumers weigh the trade-offs between sustainability and other attributes of a product, such as hedonic (e.g., esthetics) and utilitarian (e.g., functional performance) values, impacting their emotional responses and decision-making processes
Comments 3: 'moral regulation' is not clearly defined in the section 2.4. Hence H5 is not easy to follow
Response 3: Thank you for pointing this out. We appreciate your comment on setting the positioning of our paper. In response to the reviewer’s comment, we have made the following key improvements by intensifying the definition and concepts of moral regulation on Section 2.3 line 289 to 296). The added paragraph is shown as follows.
“Moral regulation refers to a process whereby individuals strive to maintain an optimal level of moral self-worth over time by balancing past actions with subsequent behaviors. This regulation involves moral cleansing, where individuals engage in moral behavior to offset prior immoral actions, and moral licensing, where past good deeds lead to less moral behavior due to a positive self-perception [101]. The relationship between moral regulation, moral cleansing, and moral licensing is characterized by a dynamic pattern of self-regulation, where individuals alternate between altruistic and selfish behaviors to maintain their perceived moral self-worth.”
Comments 4: In line 391, you would be better to use 'significant' instead of 'valid'.
Response 4: Thank you for your significant comment. Inspired by reviewer’s comment, we tend to alter ‘valid’ to ‘significant’ on the section Results of Hypothesis Testing. We elaborate more on the explanation from line 437 to 451 of the results as follows:
“The aim of this study is to examine the relationship between consumers' attitudes towards sustainability and their purchase intentions, as well as the various contextual factors that influence these intentions. Considering that sustainability involves social and environmental elements [112], this investigation examines how attitudes towards each of these dimensions, as well as the overarching attitude towards sustainability, impact purchase intention. In this study, three categories of attitudes towards sustainability (ATS) were quantified, specifically environmental, social, and holistic ATS, followed by an examination of their influence on the intent to purchase sustainable goods. We executed a multiple linear regression analysis via the software package SPSS ver.29 to evaluate how the distinct ATS can predict the purchase intention for sustainable products. Neither the attitude towards environmental sustainability (ß=.076, p>.005) nor the attitude towards social sustainability (ß=.032, p>.005) had a significant effect on purchase intention, respectively. However, the attitude towards sustainability for the single notion had a significant effect on PI (β=.032, p<.1). Therefore, we can infer that when consumers’ perception of sustainability affects their purchase intention of the sustainable product, consumers tend to portray sustainability as a whole”
Comments 5: In table 4, three-way interaction should be carefully interpreted
Response 5: We sincerely appreciate the reviewer's feedback which made our point clear. In response to the reviewer's comment, we have elaborated detailed explanation on the purpose, details in the process of the analysis on the three way interactions with ATS (Attitude towards sustainability) *Sustainability Type (Environmental/Social)*STO type(Utilitarian/Hedonic). Since this was one of the main arguments we wanted to argue therefore, we elaborated the argument and the analysis process in more details to deliver our argument to the readers more precisely and deliberately. For the detailed explanation for the analysis, we have added several points as follows: from line 526 to 556.
“We ran an ad hoc analysis on the interaction between attribute- and sustainability-type STOs in relation to PI. The analysis performed with PROCESS Macro Model 3 focused on the moderating effect of attribute- and sustainability-type STOs on the ATS–PI relationship. We initially explored the interaction effect of ATS on STOs within each division regarding PI. However, the interaction between ATS and sustainability- and attribute-type STOs does not demonstrate a significant impact on PI. Hence, sustainability- and attribute-type STOs are solely insufficient factors for valid interactions with ATS. This suggests that these factors alone are insufficient to guarantee a specific outcome in explaining the attitude–behavior gap in the sustainable product market. The results of the three-way interaction action representing the interaction effect of ATS with STOs on environmental sustainability did not reach statistical significance (p > .05), suggesting that the joint influence of ATS and environmental sustainability trade-offs (STOEs) does not have a substantial impact on purchase intention (PI). Similarly, the interaction effect of ATS with STOs on utilitarian product attributes also did not show statistical significance (p > .05), indicating that the combination of ATS and high utilitarian attributes (STOUhigh) does not notably influence PI. In contrast, the three-way interaction involving ATS*STOs on environmental sustainability*STOs on utilitarian attributes is statistically significant (p < .05), revealing a significant negative effect on PI when considering ATS, environmental sustainability trade-offs (STOEs), and high utilitarian attributes (STOUhigh) collectively. Conversely, the three-way interaction effect with ATS*STOs on environmental sustainability*STOs on hedonic attributes is not statistically significant (p > .05). The significant negative interaction effect among ATS, STOEs, and STOUhigh indicates a noticeable decrease in the purchase intention when all three factors are taken into account. This underscores a complex interplay among consumer attitudes towards sustainability, environmental trade-offs, and high utilitarian attributes, resulting in a reduced likelihood of purchase. The lack of significant interactions implies that the individual or pairwise effects of these variables do not substantially impact the purchase intention, underscoring the importance of examining the combined influences of multiple factors when evaluating consumer behavior in relation to sustainability. The results are shown in Table 6.”
Comments 6: Author Contributions: “Conceptualization; methodology; software; validation; formal analysis; 578 investigations; resources; data curation; writing—original draft preparation; writing—review and 579 editing, Je Eun Yoo." sounds the co-author is free-rider. Please also recognize the contribution of the co-author. If the current description is true, then it should be single author paper.
Response 6: Thank you for the notice. if the current description suggests that all the work was done by Je Eun Yoo alone, it does indeed imply that the co-author(s) may not have made significant contributions. For a balanced acknowledgment that recognizes the contributions of both the primary and co-authors, the statement should be revised to reflect specific contributions from everyone. The first author did involve in the research and manuscript therefore, we modified the contribution of the corresponding author as follows:
“Author Contributions: Methodology, software, validation, formal analysis, resources, data curation, writing—original draft preparation, and editing, Je Eun Yoo.
|

Reviewer 4 Report
Comments and Suggestions for Authors
An interesting paper about an interesting subject choosing the appropriate methods with some relevant results.
On the downside we have a weak implementation regarding presentation structure, readability, clarity, detail, etc. Several errors, typo and others through the paper that needs a serious English language editing because it seems that was made using a traductor like tool and additionally that the translation is not the same through the paper.
The structure is confusing because the methods used are not presented in a clear way beforehand, maybe also using in a table showing which method for each hypothesis. They are presented with low detail, sometimes presented only with the results.
The results tables don’t present all the results data and sometimes aren’t shown in the appropriate places in the article. Some results are presented through the text, others in tables, few “why” regarding the results.
The different studies explanation could have a better structure with a clear separation and different models could be present also using figures showing the pathway and the results.
More information would be useful to the reader, including the questionnaires.
Readers can ask why the figures present coffee pots alternatives A and B and through the paper they will not understand if the study includes both alternatives or why some results consider A, etc.
Discussion could have more detail with reference to results while the conclusion has less redundancy of aspects explained in the discussion, include the implications with a separate topic of Limitations and suggestions for further research.
All this adding the English language shortcomings and typo errors make it a very difficult paper to read and understand by the average reader.
Confirmation of the correctness of the references would also be needed even if I didn’t find any problem.
Considering the different aspects, we can consider that while the paper is original and with some relevant results and fit the journal scope, the results aren’t presented and interpreted in a clear and detailed way, and not all using the same presentation structure (with or without table as an example). The Discussion could have a better connection with the Results and that would also help the Conclusions.
Considering the Quality the paper needs to have a complete rewriting to eliminate errors including typo errors and to became clear and readable. The results are presented in a confusing, not structured way. Methods, data and results to be clearer with more detail as expected in a paper published in a scientific journal, also to help in their evaluation.
It could be an interesting paper with interesting conclusions for the reader if the above improvements are made, including in the English language.
Comments on the Quality of English LanguageNeeds a big improvement.
Author Response
|
Comments 1: An interesting paper about an interesting subject choosing the appropriate methods with some relevant results. |
|
Response 1: We are truly grateful for the kind and encouraging feedback you have provided on our paper. We are delighted to learn of your appreciation for the subject matter, the suitability of the chosen methods, and the significance of the results. Your constructive comments hold considerable weight with us, and we are resolute in our pursuit of improving the quality and rigor of our research.
|
|
Comments 2: On the downside we have a weak implementation regarding presentation structure, readability, clarity, detail, etc. Several errors, typo and others through the paper that needs a serious English language editing because it seems that was made using a traductor like tool and additionally that the translation is not the same through the paper. |
|
|
|
Response 2: Thank you for your valuable feedback, we appreciate it. Your concerns about the presentation, readability, clarity, and errors are noted. We apologize for the mistakes and typos in the paper, and we value your patience. We will conduct a thorough review and editing to address these issues, including English language edit. Your insights are precious, and we aim to enhance the manuscript's quality. Thank you for pointing out these matters to us.
Comments 3: The structure is confusing because the methods used are not presented in a clear way beforehand, maybe also using in a table showing which method for each hypothesis. They are presented with low detail, sometimes presented only with the results.
Response 3: We are grateful for your valuable feedback. It has come to our attention that the paper's structure may cause confusion, possibly due to the unclear presentation of the methods employed. We respectfully aim to address this issue by revising the manuscript, with the goal of clearly outlining the methods in advance, potentially using a table to illustrate the correspondence between each method and hypothesis. Additionally, gave more detailed explanations of the methods, guaranteeing they are not solely presented alongside the results. The revised details are as follows:
1) For the analysis for hypothesis 1, we have made extra detailed explanation and added the table to explain the results. For the explanation of the methods, we modified the method which was written wrong from ‘simple regression’ to ‘multiple regression’. Also, we have explained why this analysis does is needed. For the revised versions are from line 437 to 454 and written as follows:
“The aim of this study is to examine the relationship between consumers' attitudes towards sustainability and their purchase intentions, as well as the various contextual factors that influence these intentions. Considering that sustainability involves social and environmental elements [112], this investigation examines how attitudes towards each of these dimensions, as well as the overarching attitude towards sustainability, impact purchase intention. In this study, three categories of attitudes towards sustainability (ATS) were quantified, specifically environmental, social, and holistic ATS, followed by an examination of their influence on the intent to purchase sustainable goods. We executed a multiple linear regression analysis via the software package SPSS ver.29 to evaluate how the distinct ATS can predict the purchase intention for sustainable products. Neither the attitude towards environmental sustainability (ß=.076, p>.005) nor the attitude towards social sustainability (ß=.032, p>.005) had a significant effect on purchase intention, respectively. However, the attitude towards sustainability for the single notion had a significant effect on PI (β=.032, p<.1). Therefore, we can infer that when consumers’ perception of sustainability affects their purchase intention of the sustainable product, consumers tend to portray sustainability. Therefore, we can infer that the impact of ATS on PI was significant in the holistic perception of sustainability; therefore, H1 is partially supported, and the results are shown in Table 3.”
2) We also added more explanation on the analysis of hypothesis 2 for the moderation effect of sustainability-type STOs on the impact of ATS on PI. We elaborate the reason why we divided the sustainability type regarding the interaction effect of sustainability trade-off on ATS to PI. The added scripts are shown as follows:
“We then investigated how customers perceive sustainability overall when only attitude influences the purchase intention. When it comes to a trade-off with sustainability, consumers may have different reactions based on the perceived value of different sustainability attributes [113,114]; therefore, an examination was conducted to assess the moderating effect of sustainability-type trade-offs (i.e., social vs. environmental) on the impact of the ATS on PI. Specifically, a study was developed comprising four different scenarios through the classification of sustainability into two main categories—environmental and social sustainability. The aim was to investigate how the moderating impacts of the sustainability category and its strength affect customers' STO (sustainability trade-off) choices. Nevertheless, the findings did not demonstrate noteworthy results for the interplay among sustainability category, strength, and compromises.”
Comments 4: The results tables don’t present all the results data and sometimes aren’t shown in the appropriate places in the article. Some results are presented through the text, others in tables, few “why” regarding the results.
Response 4: We express gratitude for your perceptive remarks. We recognize that the exposition of the outcomes in our document requires enhancement. To tackle this issue, we have included individual tables for each examination, even if it did not demonstrate a valid impact. Nevertheless, as the reviewer pointed out that some analyses did not display the result table, hence, we have endeavored to insert table 3~5 illustrating the outcomes of each examination. Table 3 illustrates the effect of different types of attitudes towards sustainability to purchase intention. Table 4 exhibited as the effect of sustainability-type STOs on the impact of ATS on PI. Additionally, ensured that all pertinent data is thoroughly exhibited in the results tables and positioned suitably within the article. Furthermore, we incorporated more comprehensive deliberations on the rationales behind the outcomes to amplify the lucidity and profundity of our analysis. Thank you for your invaluable feedback.
Comments 5: The different studies explanation could have a better structure with a clear separation and different models could be present also using figures showing the pathway and the results.
Response 5: Thank you for your constructive feedback. We understand that the explanations of the different studies could benefit from a more organized structure. To improve this, we ensure a clear separation between the studies, and we present different models more effectively. Due to the limited number of reliable analysis outcomes, we exclusively illustrate the diagram depicting the impact of STO with type and STO with attribute type. Nevertheless, considering the noteworthy critique provided by the evaluator towards the analysis, we included a table for each analysis despite the lack of validity in the findings. We are grateful to the reviewer for pointing out this matter, which helped us in leading the readers through our research rationale using the results chart included in the document. Your recommendations are highly valued, and we are committed to diligently improving the organization and lucidity of our manuscript.
Comments 6: More information would be useful to the reader, including the questionnaires
Response 6: Thank you for your valuable feedback. We acknowledge that providing additional information would be beneficial to the reader. In response, we will include more comprehensive details, such as the questionnaires used in the experiment. our study. This will enhance the transparency and replicability of our research. We attached the questionnaire of the research. Your suggestion is greatly appreciated, and we will work to incorporate these details into the manuscript to improve its overall quality and usefulness.
Comments 7: Readers can ask why the figures present coffee pots alternatives A and B and through the paper they will not understand if the study includes both alternatives or why some results consider A, etc.
Response 7: Thank you for your feedback. We understand that the presentation of coffee pot alternatives A and B may cause confusion for readers, for there was short of explaining the rationale for choosing the experiment material. We made more detailed explanation for choosing the coffee pot as experiment material for the reader to understand more clearly. The further explanation is from line 350 to 371 about why our study choose coffee pot as the experiment material as follows:
“Our study investigated consumer sustainability trade-offs in product attributes, specifically examining whether consumers, when facing an attitude–behavior gap with respect to sustainability, tend to prioritize certain product attributes. We propose that this gap manifests differently depending on whether the product attribute is hedonic or utilitarian. Initially, we conducted the pre-test to assess consumer perceptions of the coffee pot as a product that exhibits neither purely hedonic nor utilitarian characteristics. A paired-sample t-test was revealed for a laptop, gaming console, and coffee pot. In the survey, products rated 4 or below were classified as utilitarian, while those rated above 4 were classified as hedonic. The laptop, with a mean rating of 2.33, was considered a utilitarian product, whereas the gaming console, with a mean rating of 5.75, was considered a hedonic product. However, for the coffee pot, no significant differences in consumer perception of the product as utilitarian or hedonic were observed. The results are shown as utilitarian (Mutilitarin=3.96) and hedonic (Mhedonic=4.14) attributes of the coffee pot (t=3.08, p>0.06), thus validating our selection of the coffee pot as a suitable experimental stimulus. For the main experiment, we aimed to observe how consumers perform sustainability trade-offs with the attributes of the coffee pot. We hypothesized that the attitude–behavior gap towards sustainability would influence consumers' prioritization of hedonic versus utilitarian attributes in different ways. By exploring these dynamics, our research contributes to understanding consumer behavior in sustainable product choices, particularly in contexts where products may not clearly align with traditional categorizations of hedonic or utilitarian attributes.”
Comments 8: Discussion could have more detail with reference to results while the conclusion has less redundancy of aspects explained in the discussion, include the implications with a separate topic of Limitations and suggestions for further research
Response 8: Thank you for your detailed feedback. We recognize the need to improve the discussion and conclusion sections of our manuscript. To address this, we provide a more detailed discussion with explicit references to the results, offering deeper insights and interpretations as follows:
“This investigation concentrated on various factors that impact consumers' intentions to buy sustainable products, specifically emphasizing sustainability trade-offs (STOs) and moral regulation. The findings illustrated numerous critical insights into the interaction and influence of these factors on consumer behavior. Consider H1: the attitudes towards environmental and social sustainability on purchasing sustainable products will be examined. However, the analysis indicated that the overall attitudes towards sustainability (ATSt) had a significant influence on purchase intention (PI), whereas attitudes towards environmental (ATSE) and social sustainability (ATSS) individually did not have a noteworthy effect. This suggests that consumers tend to perceive sustainability holistically rather than categorizing it into distinct dimensions. The analysis for H2 indicates that neither trade-offs in environmental (STOE) nor social sustainability (STOS) significantly moderated the impact of ATS on PI. This implies that trade-offs between different types of sustainability do not affect the impact of overall sustainability attitudes on purchasing decisions. In addition, we investigated the moderation effect of attribute-type STOs on the relationship between ATS and PI. The results show that the moderation effect of sustainability trade-offs with utilitarian (STOU) and hedonic (STOH) attributes was also not significant. None of the interactions, whether involving high or low levels of utilitarian or hedonic attributes, exhibited a significant influence (all p-values > .05). This finding implies that trade-offs involving product attributes do not independently moderate the connection between sustainability attitudes and purchase intentions. Through the ad-hoc analysis, the combined moderation effect of sustainability- and attribute-type STOs on the relationship between ATS and PI will be investigated. Specifically, the three-way interaction (ATS*STOE*STOUhigh) was significant (ß=-1.58, p<.05), indicating that the combined impact of these trade-offs significantly affects purchase intentions. This implies that consumers are particularly responsive to trade-offs involving environmental benefits and practical product features. Proposition 5: The mediating role of moral regulation in the relationship between environmental concern and purchase intention in STO conditions will be explored. Table 7 illustrates that moral regulation (MR) acts as a mediator between ATS and PI. The paths from ATS to MR (ß=.36, p<.01) and from MR to PI (ß=.47, p<.01) were significant, as was the direct path from ATS to PI (ß=.20, p<.05). This suggests that consumers' purchase decisions are significantly influenced by their moral regulation processes, aligning their sustainability attitudes with their purchase behaviors.”
In the conclusion, we reduce redundancy by avoiding repetition of points already covered in the discussion. Additionally, we will include a separate section for the implications of our findings, as well as a distinct section for limitations and suggestions for further research. Your suggestions are greatly appreciated, and we are committed to enhancing the quality and clarity of our manuscript accordingly. We have revised the STO and moral regulation part of the conclusion, focusing on the exclusivity of the perspective on the STO this research has as follows:
“Secondly, the analysis demonstrated that sustainability both in environmental (STOE) and societal sustainability (STOS) did not notably moderate the association between consumer's attitude towards sustainability and the purchase intentions (PI). When consumers engage in the STO, types of sustainability do not distinctively modify the impact of overall sustainability attitudes on purchasing choices. Similarly, for the product side, the moderating impact of sustainability compromises with utilitarian (STOU) and hedonic characteristics (STOH) on the link between ATS and PI was also insignificant. However, when the two types of STO combines, it significantly exert to the relationship of ATS to PI. This shows that in STO, both sustainability type and product attributes are important factor affecting purchase intention of the sustainable product. Even more, when we conduct the ad hoc analysis, that beside participants ATS, the interaction between two kinds of STO still impact on the purchase intention. To apply this notion in the solving the attitude behavior gap held in sustainability area, we can reconfirm that the contextual factor matters. This revalidates the previous research [12,32,39,40] positing contextual factors such as functional and symbolic product attributes, green as core value for product etc. affects the purchase intention. Nevertheless, our ad-hoc analysis uncovered a significant three-way interaction (ATS*STOE*STOUhigh), underscoring that the collective influence of these compromises substantially impacts purchase intentions. This indicates that consumers are particularly responsive to compromises involving environmental advantages and practical product attributes. This nuanced discovery emphasizes the significance of considering combined compromises in marketing sustainable goods as consumers react more favorably when practical attributes are coupled with environmental advantages. In addition, the inquiry examined the mediating role of moral regulation in the association between attitudes towards sustainability and intentions to buy. In contrast to previous studies that suggested that the level of ethics or morality in consumers influences sustainable consumption [3,18,24,106], this study confirms that the relative morality of consumers has a greater impact on their intentions to purchase sustainable products. This implies that consumers' past actions based on their morals have a stronger influence on their ethical conduct, including the purchase of sustainable products. This alignment of sustainability attitudes with buying behaviors through ethical oversight processes like moral licensing and cleansing underscores the psychological intricacy inherent in sustainable consumption. Based on the findings of this study, it can be argued that the 'conditional morality' of consumers may have a more significant impact on their decisions to purchase sustainable products.”
Comments 9: All this adding the English language shortcomings and typo errors make it a very difficult paper to read and understand by the average reader.
Response 9: Thank you for your thorough and candid feedback. We understand that the English language shortcomings and typographical errors currently make our paper difficult to read and understand for the average reader. To address these issues, we undertook a comprehensive language review and editing process to correct all grammatical errors and typos. This will enhance the readability and overall clarity of the paper. We appreciate your patience and insights, and we are committed to making the necessary improvements to ensure our manuscript is accessible and comprehensible to a broader audience.
Comments 10: Confirmation of the correctness of the references would also be needed even if I didn’t find any problem
Response 10: Thank you for your feedback. We conducted a thorough review to confirm the accuracy and correctness of all references in our manuscript, even though no specific issues were identified.
Comments 11: Considering the different aspects, we can consider that while the paper is original and with some relevant results and fit the journal scope, the results aren’t presented and interpreted in a clear and detailed way, and not all using the same presentation structure (with or without table as an example). The Discussion could have a better connection with the Results and that would also help the Conclusions.
Response 11: Thank you for your comprehensive feedback. We appreciate your acknowledgment of the originality and relevance of our results and their alignment with the journal's scope. We improved the discussion section to establish a stronger connection with the results, providing a more coherent narrative that elucidates the significance of our findings. This will help in drawing more robust and insightful conclusions. Combining more research analysis in the result, we revised the discussion part more in detail as follows:
“This investigation concentrated on various factors that impact consumers' intentions to buy sustainable products, specifically emphasizing sustainability trade-offs (STOs) and moral regulation. The findings illustrated numerous critical insights into the interaction and influence of these factors on consumer behavior. Consider H1: the attitudes towards environmental and social sustainability on purchasing sustainable products will be examined. However, the analysis indicated that the overall attitudes towards sustainability (ATSt) had a significant influence on purchase intention (PI), whereas attitudes towards environmental (ATSE) and social sustainability (ATSS) individually did not have a noteworthy effect. This suggests that consumers tend to perceive sustainability holistically rather than categorizing it into distinct dimensions. The analysis for H2 indicates that neither trade-offs in environmental (STOE) nor social sustainability (STOS) significantly moderated the impact of ATS on PI. This implies that trade-offs between different types of sustainability do not affect the impact of overall sustainability attitudes on purchasing decisions. In addition, we investigated the moderation effect of attribute-type STOs on the relationship between ATS and PI. The results show that the moderation effect of sustainability trade-offs with utilitarian (STOU) and hedonic (STOH) attributes was also not significant. None of the interactions, whether involving high or low levels of utilitarian or hedonic attributes, exhibited a significant influence (all p-values > .05). This finding implies that trade-offs involving product attributes do not independently moderate the connection between sustainability attitudes and purchase intentions. Through the ad-hoc analysis, the combined moderation effect of sustainability- and attribute-type STOs on the relationship between ATS and PI will be investigated. Specifically, the three-way interaction (ATS*STOE*STOUhigh) was significant (ß=-1.58, p<.05), indicating that the combined impact of these trade-offs significantly affects purchase intentions. This implies that consumers are particularly responsive to trade-offs involving environmental benefits and practical product features. Proposition 5: The mediating role of moral regulation in the relationship between environmental concern and purchase intention in STO conditions will be explored. Table 7 illustrates that moral regulation (MR) acts as a mediator between ATS and PI. The paths from ATS to MR (ß=.36, p<.01) and from MR to PI (ß=.47, p<.01) were significant, as was the direct path from ATS to PI (ß=.20, p<.05). This suggests that consumers' purchase decisions are significantly influenced by their moral regulation processes, aligning their sustainability attitudes with their purchase behaviors.”
Comments 12: Considering the Quality the paper needs to have a complete rewriting to eliminate errors including typo errors and to became clear and readable. The results are presented in a confusing, not structured way. Methods, data and results to be clearer with more detail as expected in a paper published in a scientific journal, also to help in their evaluation.
Response 12: We humbly appreciate your valuable feedback and the attention you have brought to the areas in our manuscript that may require improvement. A thorough revision of the paper has been undertaken to address the raised concerns. The specific steps taken are as follows: We have diligently rewritten the manuscript with the intention of enhancing its clarity and readability. This process has included a comprehensive review aimed at eliminating all typographical and grammatical errors. We restructured the results section to present the findings in a clear and organized manner. Efforts were made to ensure that we made the result table for each analysis for the presentation to align with the expected standards of a scientific journal, facilitating an easier understanding for readers to follow our research progression and outcomes. Also we tried to write the detailed descriptions incorporating the methods, data, and results sections to provide a more comprehensive understanding of our research process and findings. We have tried to improve the manuscript to ensure a logical flow of information from the introduction to the conclusion. Transitional statements have been included to guide the reader through our narrative. We believe that these revisions have significantly enhanced the quality of our manuscript. We are sincerely grateful for your constructive comments, which have played a vital role in improving our work. We eagerly anticipate your further feedback and hope that our revised manuscript will meet the esteemed standards of your journal.
Comments 13: It could be an interesting paper with interesting conclusions for the reader if the above improvements are made, including in the English language.
Response 13: We are grateful for your beneficial input. We recognize that with the proposed modifications, such as enhancements to the English language, we have completed additional English editing from a professional organization. During this process, our objective is to elevate the readability for both the audience and the evaluator. By making these changes, we aim to enhance the clarity, readability, and overall quality of the paper, making it more engaging and valuable for the readers.
|
|
4. Response to Comments on the Quality of English Language Point 1: Several errors, typo and others through the paper that needs a serious English language editing because it seems that was made using a traductor like tool and additionally that the translation is not the same through the paper. |
|
Point 2: All this adding the English language shortcomings and typo errors make it a very difficult paper to read and understand by the average reader. Point 3: It could be an interesting paper with interesting conclusions for the reader if the above improvements are made, including in the English language.
|
|
Response: Thank you for your careful comments regarding the clarity and readability of our manuscript. We sincerely apologize for any difficulties caused by the English language shortcomings and typographical errors. To enhance our limited proficiency in English, we decided to utilize the English editing assistance. Again, we appreciate your careful comments.
|

Round 2
Reviewer 1 Report
Comments and Suggestions for Authors The article presents several improvements, for its acceptance.
Author Response
|
Response to Reviewer 1 Comments
|
||
|
1. Summary |
|
|
|
We express our gratitude for the considerable dedication of your time and expertise in conducting a comprehensive evaluation of our manuscript. Your enduring mentorship is truly appreciated, and we are hopeful that these adjustments align with your expectations, thereby enhancing the overall caliber of our research.
|
||
|
2. Questions for General Evaluation |
Reviewer’s Evaluation |
Response and Revisions |
|
Is the content succinctly described and contextualized with respect to previous and present theoretical background and empirical research (if applicable) on the topic?
|
Yes |
|
|
Are the research design, questions, hypotheses and methods clearly stated?
|
Yes |
|
|
Are the arguments and discussion of findings coherent, balanced and compelling?
|
Yes |
|
|
For empirical research, are the results clearly presented?
|
Yes |
|
|
Is the article adequately referenced?? |
Yes |
|
|
Are the conclusions thoroughly supported by the results presented in the article or referenced in secondary literature?
|
Yes |
|
|
3. Point-by-point response to Comments and Suggestions for Authors
Comments 1: The article presents several improvements, for its acceptance.
Response 1: We are sincerely grateful for your kind words and acknowledgment of the enhancements made to our article. Your time and dedication to reviewing our work are greatly appreciated, and we eagerly anticipate any additional recommendations you may have.
|
||
|
4. Response to Comments on the Quality of English Language |
||
|
Nonapplicable
|
||
|
|
||
|
|
||
|
5. Additional clarifications
- Final Appreciation to the Reviewer
I am truly grateful for the insightful feedback and valuable comments you have kindly shared on our manuscript. Your suggestions have been crucial in enhancing the clarity and quality of our work. The time and effort you dedicated to reviewing our paper have been greatly appreciated, and your feedback has been invaluable in refining this manuscript. Your feedback is sincerely appreciated. Thank you very much. Sincerely,
Je Eun Yoo Corresponding Author.
|
||

Reviewer 4 Report
Comments and Suggestions for Authors
The document was globally improved in a way that can be considered for publishing. The English was improved, it's more structured and easier to read with better explanation of the concepts. The method, discussion, and conclusions improvements are good.
As minor revision:
We have two repeated figures 3 in the document, one of them without title. The figure 3 labels need to be improved, difficult to read as they are. Figure 3 could be referred in the text.
Author Response
|
Response to Reviewer 4 Comments
|
||
|
1. Summary |
|
|
|
We express our gratitude for the considerable dedication of your time and expertise in conducting a comprehensive evaluation of our manuscript. Your insights and suggestions have been incredibly helpful in shaping our revisions, which we greatly appreciate. Consequently, we have formulated a thorough point-by-point response to address your comments. Furthermore, to facilitate the assessment of these modifications, we have utilized track changes to highlight the revisions and corrections within the manuscript. Your enduring mentorship is truly appreciated, and we are hopeful that these adjustments align with your expectations, thereby enhancing the overall caliber of our research.
|
||
|
2. Questions for General Evaluation |
Reviewer’s Evaluation |
Response and Revisions |
|
Is the content succinctly described and contextualized with respect to previous and present theoretical background and empirical research (if applicable) on the topic?
|
Yes |
|
|
Are the research design, questions, hypotheses and methods clearly stated?
|
Yes |
|
|
Are the arguments and discussion of findings coherent, balanced and compelling?
|
Yes |
|
|
For empirical research, are the results clearly presented?
|
Yes |
|
|
Is the article adequately referenced?? |
Yes |
|
|
Are the conclusions thoroughly supported by the results presented in the article or referenced in secondary literature?
|
Yes |
|
|
3. Point-by-point response to Comments and Suggestions for Authors
|
||
|
Comments 1: The document was globally improved in a way that can be considered for publishing. The English was improved, it's more structured and easier to read with better explanation of the concepts. The method, discussion, and conclusions improvements are good.
|
||
|
Response 1: We are grateful for the positive feedback and recognition of the enhancements made to the manuscript. It is pleasing to note that the document has been enhanced on a global scale to a standard suitable for publication. Your suggestions have been thoughtfully addressed to enhance the clarity and readability of the English language, resulting in a more organized manuscript with improved explanations of concepts. The time and effort you dedicated to reviewing our manuscript are greatly appreciated, and we are thankful for your constructive feedback, which has significantly enriched the quality of our work.
|
||
|
Comments 2: We have two repeated figures 3 in the document, one of them without title. The figure 3 labels need to be improved, difficult to read as they are. Figure 3 could be referred in the text.
|
||
|
Response 2: Thank you for your insightful comments regarding Figure 3. We would like to clarify that the two images within Figure 3 are not duplicated but rather intentionally placed side by side to facilitate a comparison of the effects of STO (sustainability trade-offs) with different types of sustainability and attribute types. Our goal was to illustrate how the impact of STO on purchase intention may vary depending on the specific sustainability attributes being traded off. In response to your feedback, we have revised the title of Figure 3 to better reflect its purpose and have provided further explanation in the manuscript to ensure clarity for our audience as follows: ‘Figure 3. The effect of sustainability- and attribute-type STOs on PI’ to "Figure 3. The comparison of the effect of sustainability- and attribute-type STOs on PI." Also, we have improved the labels on Figure 3 to enhance readability and have clearly referenced Figure 3 within the text from line 576 to 583 as follows:
“In Figure 3, a comparative result is presented regarding the impacts of sustainability trade-offs (STO) on purchase intention across various sustainability and attribute types. The placement of the two charts side by side within the figure is deliberate, aiming to emphasize how the influence of STO to PI varies based on the specific sustainability and attributes types being traded-off. Figure 3 demonstrates that the impact of STO on purchase intention is not consistent, but rather varies according to different sustainability and attribute types." We appreciate your constructive feedback, which has helped us to improve the presentation and clarity of our findings.
|
||
|
5. Additional clarifications: Not Applicable - Final Appreciation to the Reviewer
We are deeply grateful for the insightful feedback and valuable comments you provided on our manuscript. Your suggestions have played a crucial role in improving the clarity and quality of our work. It was a great honor for us that you took the time and effort to review our paper, and we have made sure to address all your comments in the revised version. Your support means a lot to us, and we sincerely hope that the revisions meet your expectations. Sincerely,
Je Eun Yoo Corresponding Author.
|
||
